# Increasing Odontoblast-like Differentiation from Dental Pulp Stem Cells through Increase of β-Catenin/p-GSK-3β Expression by Low-Frequency Electromagnetic Field

**DOI:** 10.3390/biomedicines9081049

**Published:** 2021-08-19

**Authors:** Han-Moi Lim, Myeong-Hyun Nam, Yu-Mi Kim, Young-Kwon Seo

**Affiliations:** Department of Medical Biotechnology, Dongguk University, Goyang-si 10326, Korea; gksahl321@gmail.com (H.-M.L.); iis05047@naver.com (M.-H.N.); kjmtik@nate.com (Y.-M.K.)

**Keywords:** human dental pulp stem cells, odontoblasts, pulsed electromagnetic field, odontogenesis, GSK-3β, β-catenin

## Abstract

Odontoblasts produce proteins that form the dentinal extracellular matrix, which can protect the dental pulp from external stimuli and is required for tooth regeneration. This study showed that a pulsed electromagnetic field (PEMF) can regulate cell metabolism and induce cell differentiation. This study determined the frequency of PEMF that is effective for odontoblast differentiation. Human dental pulp stem cells (hDPSCs) were cultured in odontoblast differentiation medium containing dexamethasone, BMP2, TGF-β1, and FGF-2, and then exposed to 10 mT intensity of PEMF at 40, 60, 70, and 150 Hz for 15 min/day. The MTT assay, LDH assay, flow cytometry, protein and gene expression, and immunofluorescence were performed to check if hDPSCs differentiated into odontoblast-like cells. The hDPSCs showed frequency-dependent differences in protein and gene expression. The mesenchymal stem cell markers were reduced to a greater extent at 60 and 70 Hz than at other frequencies, and odontoblast-related markers, particularly β-catenin, p-GSK-3β, and p-p38, were increased at 60 and 70 Hz. Exposure to 10 mT intensity of PEMF at 70 Hz influenced the differentiation of hDPSCs considerably. Taken together, PEMF treatment can promote differentiation of hDPSCs into odontoblast-like cells by increasing p-GSK-3β and β-catenin expression.

## 1. Introduction

Human dental pulp stem cells (hDPSCs) are located in teeth [1,2], where they can differentiate into specific cell types, similar to other mesenchymal stem cells (MSCs) [3,4,5,6]. hDPSCs can also differentiate into cells called odontoblasts that form a single-cell layer along the dental pulp–dentin border [7]. Odontoblasts secrete the extracellular matrix components of dentin which act as barriers to dental pulp protection from external stimuli. These specialized cells produce proteins that form the dental pulp extracellular matrix, which is comprised of dentin matrix acidic phosphoprotein 1 (DMP-1) and dentin sialophosphoprotein (DSPP) [8,9,10,11]. DMP-1 and DSPP are considered typical markers of odontogenesis [12,13] and reportedly play a role in mineralization [14,15]. Additionally, DMP-1 plays a key role in odontogenesis, and its expression is essential for late dentinogenesis development [8].

Substantial research has been conducted on odontoblast differentiation. A comparative study revealed the differentiation of DPSCs, bone marrow-derived mesenchymal stem cells (BM-MSCs), and adipose-derived MSCs into odontoblast-like cells using a differentiation medium [16]. Most researchers performed studies using growth factors and cytokines. Kidwai et al. [17] and Ozeki et al. [18] differentiated embryonic stem cells into odontoblast-like cells using bone morphogenetic protein-4. Jiang et al. induced the differentiation of cranial neural crest cells into odontoblast-like cells by fibroblast growth factor 8 and adult extracellular matrix proteins [17]. In another study, BM-MSCs and amniotic MSCs were differentiated into odontoblast-like cells using a porous scaffold and mechanical stimulation [19]. Those authors applied 19.6 kPa of mechanical compression onto the microporous scaffold for nine hours, which increased the expression of odontoblast-related markers [18]. Likewise, in a study by Liao et al., mechanical tension induced odontogenic-like differentiation of hDPSCs through the STAT3 signaling pathway by downregulating sclerostin expression and accelerated senescence of dental pulp cells [20]. Another study promoted odontogenic-like differentiation of BM-MSCs by co-culturing with oral epithelial cells [21]. The differentiation-induced cells showed increased expression of DMP-1 and DSPP, and when transplanted in a model rat with dentin defect, a dentin-like structure was regenerated [19].

Electromagnetic fields (EMFs) affect cell activity, proliferation, and differentiation by affecting cell membranes or intracellular proteins, such as ion channels [21,22]. The cell differentiation effects of EMFs have been widely reported, including the differentiation of MSCs into nerve-, bone-, and joint-related specialized cells [23,24,25]. Since Basset et al. first introduced EMFs for bone fracture therapy in 1974 [26], other investigators have used EMFs to increase osteogenic activity and differentiation of bone cells [27,28,29].

Martino et al. revealed a dramatic increase in mineral nodule formation and alkaline phosphatase (ALP) activity of SaOS-2 cells exposed to 0.9 mT and 15 Hz of EMF for four hours per day [29]. Jazayeri et al. showed that 15 Hz and 0.2 mT of EMF exposure for six hours per day induced differentiation of rat MSCs to osteoblast-like cells [30]. Another researcher exposed (24 h/day) alveolar bone-derived MSCs to various frequencies (10–100 Hz) of EMF (0.6 mT) and noticed a high increase in the proliferation, mineralization, ALP, vinculin, vimentin, and calmodulin expression at 50 Hz [31]. By contrast, 15 Hz and low intensity (0.1 mT) of EMF exposure for eight hours per day did not affect the extracellular matrix synthesis of mouse osteoblasts, decreased ALP activity and the receptor activator of nuclear factor-kappa B ligand (RANKL) expression, but increased osteoblastic proliferation [32].

Analysis of the related research suggests that the effect of EMF can be regulated by controlling the intensity, frequency, and exposure time, affecting the proliferation and differentiation of various cells. Thus, we hypothesized that odontoblastic differentiation would be affected by exposure to high-intensity (10 mT) short-time (15 min/day) PEMF. We verified the effects of PEMF by analyzing the gene and protein expression levels of hDPSCs exposed to PEMF at different frequencies to find changes indicative of abilities of DPSC to differentiate into odontoblast-like cells.

## 2. Materials and Methods

### 2.1. Preparation and Culture of hDPSCs

hDPSCs were purchased from Lonza (PT-5025; Basel, Switzerland). The cells were cultured in a growth medium; a Dulbecco’s modified Eagle’s medium (DMEM) with low glucose (Welgene, Daejeon, Korea) supplemented with 10% D-glucose (Welgene), fetal bovine serum (FBS; BioWhittaker^TM^, Cambrex Biosciences, Walkerville, MD, USA), and penicillin–streptomycin (Welgene). The final concentration of medium constituents was 0.09% D-glucose, 10% FBS, and 1% penicillin–streptomycin. The cells were incubated at 37 °C in a humidified atmosphere with 5% CO_2_.

### 2.2. Odontoblast-Like Differentiation of hDPSCs

To induce odontoblast-like differentiation, hDPSCs were treated with odontoblastic differentiation medium; growth medium with 10 nM dexamethasone, 50 ng/mL BMP2, 20 ng/mL transforming growth factor-β1, and 5 ng/mL fibroblast growth factor-2 (all from Sigma-Aldrich, St. Louis, MO, USA), for 3–10 days with or without PEMF [33]. Control cultures were placed elsewhere to avoid exposure to PEMF and were incubated with the same medium as described for the experimental cultures.

### 2.3. PEMF

We used Helmholtz coils to generate PEMF (Figure 1). The stimulus form was a pulse, and the stimulation intensity was 10 mT. The PEMF frequencies were 40, 60, 70, and 150 Hz. The PEMF device was placed in an incubator at 37 °C with 5% CO_2_. The cells were exposed to PEMF for 15 min/day for 3 days.

### 2.4. Cell Viability

To evaluate cell viability, MTT analysis was performed after 3 days of culturing. Cell activity was assessed using the 3-(3,4-dimethylthiazol-2-yl)-2,5-diphenyltetrazolum bromide assay (MTT assay). Three days after the cells (5 × 10^4^) were seeded in 35-mm dishes, each dish was treated with an MTT solution (5 mg/mL), followed by incubation at 37 °C for 90 min. After incubation, the solution was replaced with dimethyl sulfoxide (Sigma–Aldrich) to dissolve the formazan. The absorbance of the resulting solution was measured at 570 nm.

### 2.5. Cytotoxicity Assay

Cell cytotoxicity was measured by LDH assay (LDH assay kit, Takara Bio, Inc., Shiga, Japan) in preservation media, according to the manufacturer’s protocol. MTT analysis was performed after 3 days of culturing.

### 2.6. Cell Surface Antigen Analysis by FACS

To analyze differentiation, hDPSCs were cultured in the same condition as described above (Section 2.2 and Section 2.3) for 10 days. The cell surface markers present on hDPSCs were analyzed by flow cytometry. Briefly, hDPSCs were incubated with MSC-positive antibodies (CD73, CD105, and CD146, BioLegend, San Diego, CA, USA) for confirming differentiation. Cell suspensions in 5% FBS without antibodies served as controls. The cells were washed three times with 5% FBS to remove unbound antibodies and then resuspended with 200 μL of 5% FBS. The cells were sorted using a flow cytometer (CytoFLEX, Beckman Coulter, Brea, CA, USA) and analyzed using CytExpert software (Beckman Coulter).

### 2.7. Western Blotting

To evaluate protein expression, Western blotting was performed after 3 days of culturing. Whole-cell proteins were extracted as previously described [34]. Protein concentrations were determined by a bicinchoninic acid assay. Herein, 10 μg of protein was separated at 90 V on 10% SDS–PAGE for 120 min. The separated proteins were transferred onto a nitrocellulose membrane at 120 V for 90 min and then incubated with primary antibodies against ALP, Runx2, DMP-1, DSPP, β-catenin, GSK-3β, p-GSK-3β, p38, and p-p38 (all from Abcam). The membranes were incubated with 2 μg of anti-mouse and anti-rabbit secondary antibodies in 5% skimmed milk in TBS buffer for 1 h and applied to an enhanced chemiluminescence (ECL) solution for 1 min. Then, autoradiography was performed using a ChemiDoc XRS+ Imaging System (Bio-Rad, Hercules, CA, USA). β-Actin served as an internal protein control.

### 2.8. RT-PCR and Real-Time PCR Analysis

To evaluate mRNA expression, RT-PCR analysis was performed after 3 days of culturing. TRIzol (Invitrogen, Waltham, MA, USA) was used to isolate total RNA from the cells. After adding 500 µL of TRIzol reagent to a 60-mm culture dish and collecting it in an e-tube, 200 µL of chloroform was added into the e-tube, and the solution was vortexed and incubated for 5 min. After centrifugation (13,475× *g*, 4 °C, 15 min), the supernatant was transferred to a new tube, and 500 µL of isopropanol was added. After a 10-min incubation and another centrifugation step (18,340× *g*, 4 °C, 10 min), the solution was discarded. The pellet was washed with 1 mL of 75% ethanol and centrifuged (7580× *g*, 4 °C, 5 min). The solution was removed, and the pellet was air-dried. The pellet was dissolved in 20 µL of RNase-free water (Welgene) and incubated on ice for 10 min. Total RNA was determined using a NanoDrop spectrophotometer (Thermo Fisher Scientific, Waltham, MA, USA).

For the RT-PCR analysis, reverse transcriptase reactions were used to synthesize cDNA from 3 µg of total RNA using an Advantage RT-PCR kit (Clontech Laboratories, Inc., Palo Alto, CA, USA) by following the manufacturer’s protocols (Table 1). Test gene expressions were normalized against β-actin in each sample.

Additionally, for the real-time PCR analysis, reverse transcriptase reactions were used to synthesize cDNA from 2 µg of total RNA using a dyne RT Dry MIX (Dyne Bio, Gyeonggi-do, Korea) by following the manufacturer’s protocols. qPCR was performed using a TB Green^®^ Premix Ex Taq™ (Tli RNaseH Plus) and a StepOnePlus Real-Time PCR System (Applied Biosystems, Waltham, MA, USA) with 60 cycles of denaturation at 95 °C for 5 s, annealing at 60 °C for 25 s, and amplification at 72 °C for 30 s. Test gene expressions were normalized against GAPDH in each sample.

### 2.9. Immunofluorescence

To evaluate antigen expression, immunofluorescence staining was performed after 5 days of culturing. hDPSCs were grown on round, sterilized glass cover slips (12 mm). The cells were characterized for the markers of differentiated hDPSCs (DMP-1 and β-catenin; Abcam). Briefly, the cells were washed with cold Dulbecco’s phosphate-buffered saline (PBS), fixed in 4% paraformaldehyde for 10 min, and washed twice in PBS with 0.1% Triton X-100 (PBS-T) for 5 min. Afterward, the cells were incubated with primary antibodies diluted in 1% bovine serum albumin (primary antibodies; DMP-1 and β-catenin, Abcam) overnight and then with appropriate secondary antibodies for 2 h. After washing twice with PBS-T, the nuclei were stained with DAPI (0.4 µg/mL) at room temperature for 2 min in the dark. The cells were washed twice with PBS-T, mounted on clean slide glasses with Mount Fluor (BioCyc, Potsdam, Germany), and stored at 4 °C. Representative images were captured using a Nikon Eclipse Ti microscope.

### 2.10. Von Kossa Staining

To evaluate calcium deposition, von Kossa staining was performed after 10 days of culturing. The cells were assessed using a 5% silver nitrate (Sigma-Aldrich) solution under ultraviolet light for 20 min, followed by a 5% sodium thiosulphate (Sigma-Aldrich) solution for 5 min, and then counterstained with a nuclear fast red solution (Sigma-Aldrich) for 10 min. The mineral was stained black.

### 2.11. Statistical Analysis

Data are expressed as the mean ± SEM of three independent experiments. One-way analysis of variance (ANOVA) followed by Tukey–Kramer multiple comparisons test was performed with GraphPad Prism (La Kolla, CA, USA). Mean differences were considered significant at *p* < 0.05 (* *p* < 0.05, ** *p* < 0.01, and *** *p* < 0.005). Graphical representations were created using SigmaPlot (Systat Software, Inc., San Jose, CA, USA). All experiments were performed in triplicate.

## 3. Results

### 3.1. Non-Cytotoxicity of PEMF Exposure

We exposed hDPSCs to various frequencies (40, 60, 70, and 150 Hz) of PEMF at an intensity of 10 mT and performed lactate dehydrogenase (LDH) and mitochondrial activity (MTT) assays. Figure 2A shows the morphologies of hDPSCs in the control group and the PEMF-exposed groups after three days. All control and PEMF-treated groups, irrespective of the PEMF frequency, did not show apoptosis or necrosis (Figure 2A), and there were no differences in mitochondrial activities between the groups in the MTT assay (Figure 2(Ba)). Furthermore, the absence of significant differences in the amount of LDH released between the groups exposed to PEMF and the controls confirmed that PEMF did not induce cellular stress (Figure 2(Bb)).

### 3.2. hDPSCs Were Differentiated by Media and PEMF

Cell differentiation was evaluated by fluorescence-activated cell sorter (FACS) analysis of MSC-related cell surface proteins because hDPSCs express cell surface proteins similar to those of MSCs [35,36]. Three known MSC markers, namely CD73, CD105, and CD146, were used in FACS analysis to determine whether PEMF alters hDPSC surface antigen expression (Figure 3). The cell surface proteins were expressed in over 95% of the undifferentiated hDPSCs (data not shown). The results of cultures after five days showed the expression of 61.55 ± 1.57% and 61.29 ± 0.32% CD73, 29.24 ± 6.51% and 29.33 ± 4.34% CD105, and 32.41 ± 2.16% and 30.19 ± 3.24% CD146 at 60 and 70 Hz, respectively, and 77.21 ± 1.47% CD73, 43.46 ± 1.09% CD105, and 40.38 ± 0.53% CD146 expression in the control group (Figure 2A). All surface antigens were decreased in all groups after 10 days, particularly the cells exposed to 60 and 70 Hz PEMF (Figure 3B), which indicates that some frequencies of PEMF may cause changes in hDPSC surface antigen expression. The FACS percentages are shown in Table 2.

### 3.3. PEMF Exposure-Induced High Expression of Odontoblast-Related Molecules

Markers related to odontoblastic differentiation were generally increased in the cells exposed to PEMF (Figure 4). The frequency of 70 Hz showed a particularly high expression of these markers (1.4-fold in runt-related transcription factor 2 (Runx2), 6.4-fold in DMP-1, and 1.6-fold in DSPP). Similarly, the cells exposed to 60 Hz PEMF showed high expression as well (3.3-fold in ALP, 1.3-fold in Runx2, 4.1-fold in DMP-1, and 1.3-fold in DSPP). Western blotting results showed that gene expressions of *DMP-1* and *DSPP* were generally higher at 60 and 70 Hz than those in the other groups, and the cells exposed to 60 Hz PEMF expressed the highest *ALP* levels.

As shown in Figure 5A, all odontogenic genes (bone morphogenetic protein (*BMP*) *2*, *ALP*, *Runx2*, osteomodulin (*OMD*), *DMP-1*, and *DSPP*) were expressed more in PEMF-exposed cells than in the controls based on reverse transcriptase-polymer chain reaction (RT-PCR) analysis. BMP, OMD, Runx2, and DSPP expression had slight but not statistically significant differences in frequency. However, the frequency of 60 and 70 Hz showed a high expression in ALP and DMP-1 markers compared to those of the other frequency, and the difference was statistically significant.

However, real-time PCR results differ from RT-PCR. For BMP, OMD, ALP, and Runx2 expression, there was a slight increase in the EMP-exposure groups compared to the non-exposure group, but there were no statistical differences. In particular, DSPP showed a slight decrease in expression compared to that in non-exposure group groups but increased expression at 150 Hz (Figure 5B). As a result, no statistical differences were observed in the expression of odontoblast-related mRNA by EMF-exposure.

### 3.4. DMP-1 and β-Catenin Increased in Nucleus and Cytoplasm of Cells Exposed to PEMF of Certain Frequency

hDPSCs were treated with PEMF at various frequencies, and the expression of DMP-1 and β-catenin was measured via immunofluorescence staining (Figure 6). Beta-catenin was upregulated at 60 and 70 Hz, and DMP-1 also showed increased expression at these frequencies. In particular, the expression of both proteins increased noticeably at 70 Hz in merged images of DMP-1, β-catenin, and 4′,6-diamidino-2-phenylindole (DAPI).

### 3.5. PEMF Exposure at Certain Frequencies Promotes Odontoblast-Related Proteins through Phosphorylation of Glycogen Synthase Kinase-3 Beta (GSK-3β)/β-Catenin

We aimed to determine whether GSK-3β/β-catenin signaling was involved in the differentiation of hDPSCs and the expression of odontoblast-related proteins caused by PEMF. As seen in Figure 7, PEMF increased the overall phosphorylation of GSK-3β/β-catenin significantly. Furthermore, phosphorylated-GSK-3β (p-GSK-3β (Ser9)) and phosphorylated-p38 (p-p38) levels increased markedly by approximately 1.9- and 1.8-fold after treatment with 70 Hz PEMF, respectively. The expression of β-catenin, a transcription coactivator known to affect odontoblast differentiation in GSK-3β/β-catenin signaling, increased by about 9.5-fold at 60 Hz and 15.0-fold at 70 Hz. These data collectively showed that PEMF induced hDPSC differentiation and increased odontoblast-related protein expression levels via the pathway mediated by GSK-3β/β-catenin signaling.

### 3.6. Increased Calcium Deposition by PEMF Exposure

The von Kossa staining performed for the evaluation of the calcium deposition of the hDPSC during differentiation (Figure 8). In the growth media culture group, no mineral deposited on the cell surface was observed at all. The EMF-exposed groups exhibited a small amount of matrix mineralization, while the non-exposed (control) group exhibited little matrix mineralization compared with the other groups on Day 10. Not enough mineral deposition was observed due to the short culture period of the cells. However, mineral deposition was formed along the cytoplasm and was observed on the cell surface. At frequency above 60 Hz, the minor deposition difference was not identified.

## 4. Discussion

EMF is known to be non-invasive and non-toxic and exerts diverse biological effects on the activity, growth, and differentiation of various cell types [37,38,39,40]. The purpose of this study was to identify the effects of various frequencies of PEMF on hDPSCs and also determine the optimal condition that would promote their differentiation into odontoblast-like cells. To our knowledge, this is the first study on the use of PEMF for odontogenesis.

After three days of incubation with PEMF exposure, cell activity remained similar at all frequencies without affecting mitochondrial activities (Figure 2(Ba)). We observed cell morphology and performed the LDH assay to check for cytotoxicity. No morphological changes in the cell membrane and no cytotoxicity, such as vacuole or signs of apoptosis, were observed following PEMF exposure at any frequency (Figure 2(Bb)).

Some investigators have reported that high-frequency magnetic exposure can induce mitochondrial DNA damage, reactive oxygen species, and cell apoptosis. For example, EMF of 8 mT (50 Hz) and long time (eight hours per day for eight weeks) induced an increase in mitochondrial damage and ultra-structure changes [41,42], and EMF of 10 mT (50 Hz) and long time (24 hours per day for three weeks) increased stress and abnormal behavior in rat models [43]. However, another study showed that EMF exposure of 16 mT (15 Hz) and short time (eight hours a day for 24 days) increased osteoblastic differentiation of MSCs [44]. These findings indicate that the effect of EMF on cells or animals can be regulated by the applied frequency, intensity, and exposure time [45]. According to the results of the LDH and MTT assays in the current work, although the intensity (10 mT) was high, the short exposure (15 min/day) time showed that PEMF did not cause cell damage and stress.

hDPSCs are known as adult stem cells. Both their features and the expression of their cell surface proteins are similar to those of MSCs [36]. Based on this, the cells were analyzed using FACS with CD73, CD105, and CD146 as the representative surface markers of MSCs because these markers decrease when MSCs differentiate into specific cell types [46,47,48,49]. On day five of culturing, the cells exposed to 60 and 70 Hz PEMF expressed all three MSC markers at levels about 10% lower relative to those of the control (Figure 3A). On day 10 of culturing, most of these levels had declined to below 10%, indicating the differentiation of hDPSCs (Figure 3B). In particular, the expression levels were lower in the cells exposed to 60 and 70 Hz PEMF than in the control group. Consequently, the expression levels of the cell surface markers decreased regardless of whether the cells were exposed or not exposed to PEMF, and these results indicate cell differentiation.

Figure 4 shows the protein expressions in each experimental group. ALP is a critical marker of osteo/odontoblastic differentiation and associated with mineralization [50,51,52]. Here, ALP showed high expression at 60 Hz. Runx2 was more expressed in the hDPSCs exposed to 60 and 70 Hz PEMF compared with the control group. Furthermore, the increase in the expression of Runx2, which is related to the repair of dental pulp damage, regulated the odontoblastic differentiation of hDPSCs [53,54]. DMP-1 and DSPP play key roles in dentin mineralization and maturation as major markers of odontoblasts and have been previously used to determine odontoblast-like differentiation [12,14,55,56,57]. In our study, DMP-1 was significantly expressed at 70 Hz and was high at other frequencies as well. Similarly, DSPP showed high expression at 60 and 70 Hz. Immunofluorescence showed that the expression of DMP-1 in both cytoplasm and nucleus was high at 70 Hz (Figure 6).

Through RT-PCR, we analyzed the expression of odontoblastic differentiation-related genes (Figure 5). *BMP2* is an important regulator of the differentiation process because it promotes odontoblastic differentiation and supports forming mineralized nodules [58,59,60]. *BMP2* gene expression was increased in the cells that were exposed to PEMF at 60 Hz. *OMD* was also expressed in cells exposed to any frequency. Interestingly, Lin et al. demonstrated that knockdown of *OMD* reduced mineralized nodule formation and inhibited osteo/odontoblastic differentiation [61]. Regarding protein expressions, *ALP*, *Runx2*, *DMP-1*, and *DSPP* showed increased expression in hDPSCs exposed to PEMF at any frequency. The effect of odontoblast-related protein and gene expression may lead to the differentiation of hDPSCs exposed to PEMF.

Figure 7 shows the significantly increased expression of β-catenin at 70 Hz. It was also verified that the phosphorylation of GSK-3β and p38 associated with the activation of β-catenin increased at 70 Hz (Figure 7). β-Catenin regulates the proliferation and differentiation of various cells, which is exemplified by its roles in osteogenesis and hair growth [62,63]. Han et al. induced tooth regeneration by odontoblastic differentiation in their experiments with rat teeth and cells, and confirmed by immunofluorescence that catenin protein and gene expression increased [53]. It is well-known that active β-catenin is transferred from the cytoplasm to the nucleus and plays a crucial role in dentinogenesis due to the proliferation and odontoblastic differentiation of hDPSCs [64]. Similar to osteogenesis, odontoblast differentiation also occurs through the GSK-3β/β-catenin pathway. In vivo, GSK-3β suppression temporarily increased bone formation, weakened bone degradation, and inhibited osteolysis induced by mechanical instability of the implants [65]. Additionally, the canonical Wnt/β-catenin pathway inhibits GSK-3β activity, which leads to the accumulation of β-catenin in the cytosol and its subsequent translocation to the nucleus [66]. Odontoblastic differentiation was promoted through the WNT, p38 mitogen-activated protein kinase (MAPK), and BMP signaling pathways [67] and suppressed by the repression of p-GSK-3β and β-catenin [68]. Disruption of p38 can affect the integration factors required for various signal inputs. In vivo, p38 deletion inhibits osteoblast terminal differentiation and the appearance of osteocytes, which directly affects bone composition and maintenance [69]. Yun et al. differentiated hDPSCs into odontoblast-like cells using a differentiation medium on a nanofiber scaffold and confirmed to the activity of p38 MAPK signaling by analyzing the protein expression. The results confirmed that p38 MAPK signaling was activated, and p-GSK-3β was increased when hDPSCs were differentiated into odontoblast-like cells [70]. Combining the known information with our findings, we can conclude that PEMF modulates the differentiation of hDPSCs into odontoblasts through the p38 MAPK signaling pathway and the GSK-3β/β-catenin pathway.

## 5. Conclusions

Based on the results, the characteristics of odontoblasts were well-expressed after exposure to 70 Hz PEMF for 15 min. Consequently, we have demonstrated for the first time that hDPSCs can effectively differentiate into odontoblast-like cells when exposed to PEMF. Short-term PEMF exposure methods have suggested the possibility of incorporating PEMF therapy into future dentin regeneration. In addition, non-invasive and non-toxic PEMF will provide new methods for periodontal therapy.

## Figures and Tables

**Figure 1 biomedicines-09-01049-f001:**
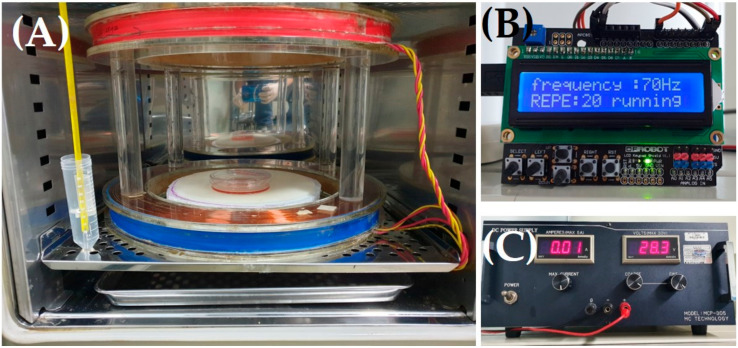
Image of the PEMF stimulation device. Helmholtz coils (**A**), a function generator set to 70 Hz ((**B**); 40, 60, 70, and 150 Hz), and a power supply (**C**) were used. Coils were placed in an incubator (37 °C, 5% CO_2_). PEMF; pulsed electromagnetic field.

**Figure 2 biomedicines-09-01049-f002:**
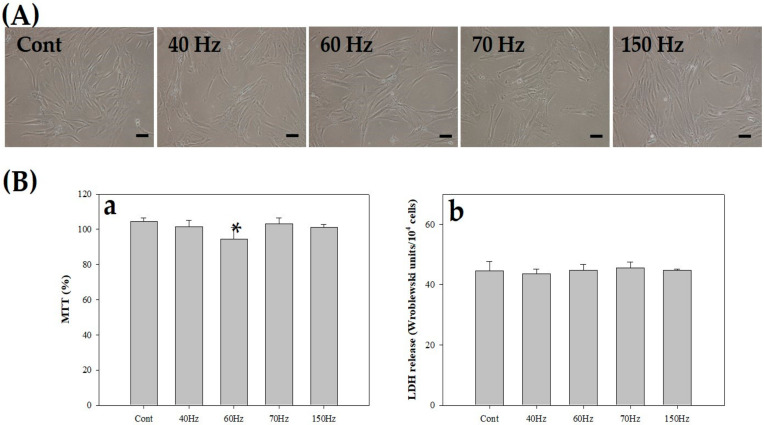
(**A**) Morphologies of human dental pulp stem cells (hDPSCs) after PEMF exposure for 3 days (10 mT). All groups were incubated with the same medium—odontoblastic differentiation medium. Each group had different frequency conditions (40, 60, 70, and 150 Hz). Scale bar = 100 μm. (**B**) Cell viability (**a**; MTT assay) and stress (**b**; LDH assay) of hDPSCs at 3 days. * *p* < 0.05 (compared with the control). PEMF; pulsed electromagnetic field; MTT; 3-(3,4-dimethylthiazol-2-yl)-2,5-diphenyltetrazolum bromide, LDH; lactate dehydrogenase. Cont = control (no PEMF treatment).

**Figure 3 biomedicines-09-01049-f003:**
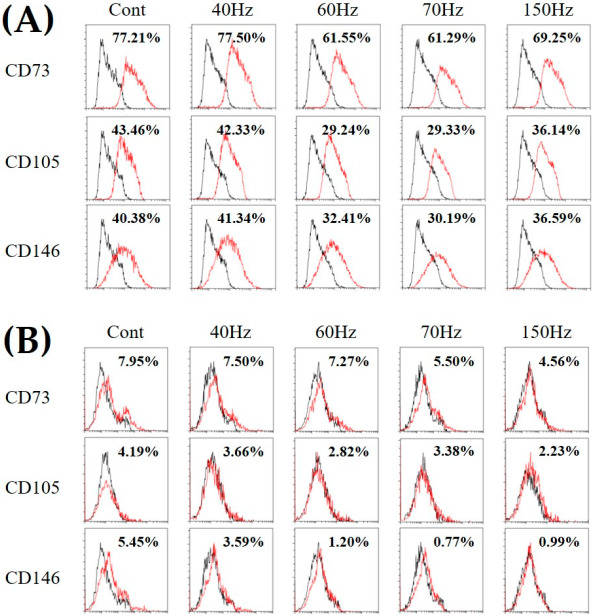
Fluorescence-activated cell sorter analysis of the surface markers CD73, CD105, and CD146 after PEMF exposure for 5 days (**A**) and 10 days (**B**). hDPSCs were labeled with phosphatidylethanolamine-conjugated antibodies and then analyzed in a flow cytometer. hDPSCs; human dental pulp stem cells. Cont = control (no PEMF treatment). Black line: FITC- or PE-conjugated IgG1, Red line: FITC- or PE-conjugated anti-body.

**Figure 4 biomedicines-09-01049-f004:**
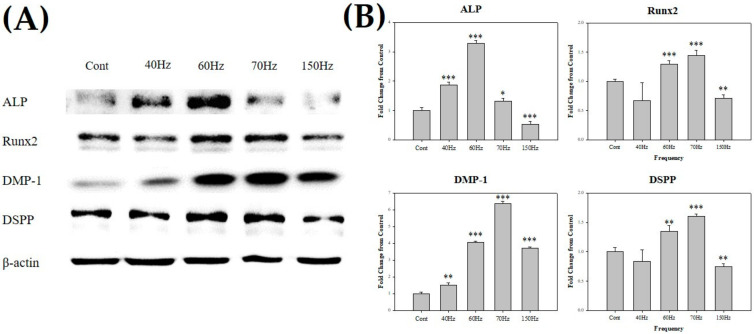
(**A**) Western blot of ALP, Runx2, DMP-1, DSPP, and β-actin after exposure to 10 mT PEMF for 3 days. (**B**) All markers were increased in the cells exposed to PEMF. Each target protein was normalized to β-actin. Notably, most markers were expressed at 60 and 70 Hz. * *p* < 0.05, ** *p* < 0.01, *** *p* < 0.005 (compared with the control). ALP; alkaline phosphatase, Runx2; runt-related transcription factor 2, DMP-1; dentin matrix acidic phosphoprotein 1, DSPP; dentin sialophosphoprotein, ACTB; β-actin, PEMF; pulsed electromagnetic field. Cont = control (no PEMF treatment).

**Figure 5 biomedicines-09-01049-f005:**
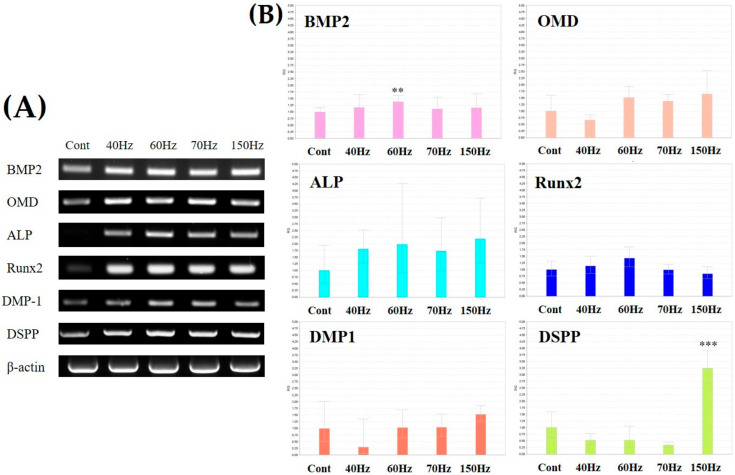
(**A**) RT-PCR analysis of *BMP2*, *ALP*, *Runx2*, *OMD*, *DMP-1*, *DSPP*, and *ACTB* after PEMF exposure for 3 days. (**B**) Real time-PCR analysis for odontoblast-related marker. All markers were increased in the cells exposed to PEMF. ** *p* < 0.01, *** *p* < 0.005 (compared with the control). *BMP2*; bone morphogenetic protein 2, *OMD*; osteomodulin, *ALP*; alkaline phosphatase, *Runx2*; runt-related transcription factor 2, *DMP-1*; dentin matrix acidic phosphoprotein 1, *DSPP*; dentin sialophosphoprotein, *ACTB*; β-actin, PEMF; pulsed electromagnetic field. Cont = control (no PEMF treatment).

**Figure 6 biomedicines-09-01049-f006:**
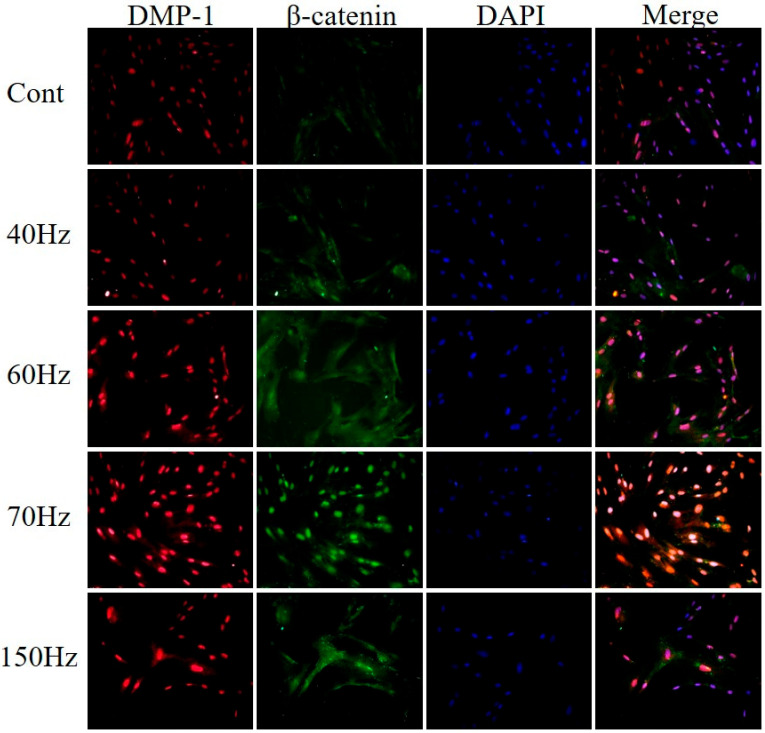
Effects of PEMF exposure on hDPSC odontogenesis as determined by immunofluorescence of DMP-1 and β-catenin at 5 days. DMP-1 and β-catenin expressions were visibly increased in PEMF-stimulated cells. PEMF; pulsed electromagnetic field, hDPSC; human dental pulp stem cell, DMP-1; dentin matrix acidic phosphoprotein 1. Cont = control (no PEMF treatment).

**Figure 7 biomedicines-09-01049-f007:**
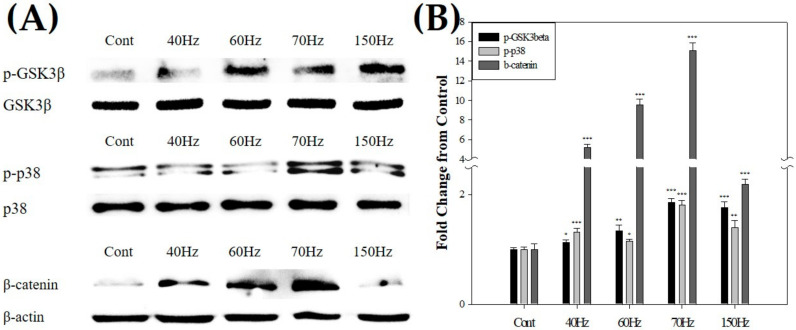
(**A**) Western blot of β-catenin, p-GSK3β, and p-p38 after 10 mT PEMF exposure for 3 days. (**B**) All markers were increased in the cells exposed to PEMF. PEMF-exposed groups had higher expression than the control group. * *p* < 0.05, ** *p* < 0.01, *** *p* < 0.005 (compared with the control). p-GSK3β; phosphorylated-glycogen synthase kinase-3 beta, p-p38; phosphorylated-p38, PEMF; pulsed electromagnetic field. Cont = control (no PEMF treatment).

**Figure 8 biomedicines-09-01049-f008:**
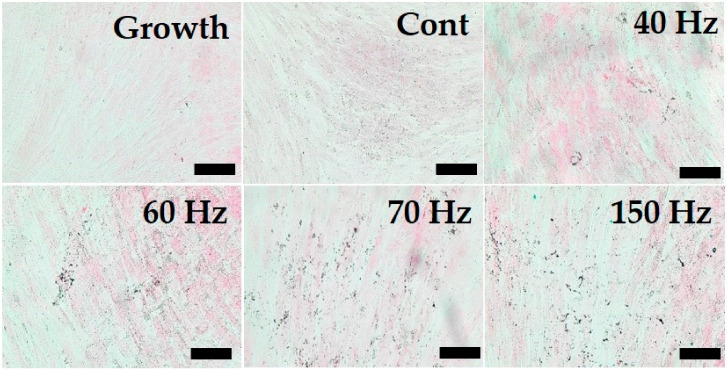
Effects of PEMF exposure on calcium deposition as determined by von Kossa staining at 10 days. Mineral detections were increased in PEMF-stimulated cells. PEMF; pulsed electromagnetic field, hDPSC; human dental pulp stem cell, Cont = control (no PEMF treatment). Growth = growth media (no PEMF treatment). Original magnification: ×200, bar = 100 μm.

**Table 1 biomedicines-09-01049-t001:** PCR primer sequences. *BMP2*; bone morphogenetic protein 2, *ALP*; alkaline phosphatase, *Runx2*; runt-related transcription factor 2, *OMD*; osteomodulin, *DMP-1*; dentin matrix acidic phosphoprotein 1, *DSPP*; dentin sialophosphoprotein.

Gene	Forward (5′-3′)	Reverse(5′-3′)	Temp (°C)	Cycles	Base Pair (bp)
β-actin	GTGATGGTGGGCATGGGTCA	GCCGGACTCGTCATACTCCT	58.0	37	972
BMP2	GTCCAGCTGTAAGAGACACC	GTACTAGCGACACCCACAAC	58.5	28	316
ALP	ATCTCGTTGTCTGAGTACCAGTCC	TGGAGCTTCAGAAGCTCAACACCA	59.0	32	454
Runx2	ACAGTAGATGGACCTCGGGA	ATACTGGGATGAGGAATGCG	55.0	30	113
OMD	GCTATGGATGGGCTAGTAAAC	GGGATGTCTTGTAGTTTGTTGTG	58.5	36	278
DMP-1	GAGTGGCTTCATTGGGCATAG	GACTCACTGCTCTCCAAGGG	60.0	37	260
DSPP	GGAATGGCTCTAAGTGGGCA	CTCATTGTGACCTGCATCGC	60.0	37	284

**Table 2 biomedicines-09-01049-t002:** FACS analysis of mesenchymal stem cell markers. FACS data percentage—upper values in each row; 5 days of PEMF exposure, lower values in each row; 10 days of PEMF exposure. Cont = control (no PEMF treatment).

Markers%		Cont	40 Hz	60 Hz	70 Hz	150 Hz
CD73	5 D	77.21 ± 1.47	77.50 ± 4.72	61.55 ± 1.57 ***	61.29 ± 0.32 ***	69.25 ± 2.71 *
	10 D	7.95 ± 0.34	7.50 ± 2.14	7.27 ± 3.29	5.50 ± 1.97	4.56 ± 2.12
CD105	5 D	43.46 ± 1.09	42.33 ± 2.87	29.24 ± 6.51 *	29.33 ± 4.34 *	36.14 ± 3.76 *
	10 D	4.19 ± 0.49	3.66 ± 1.22	2.82 ± 1.07	3.38 ± 0.92	2.23 ± 1.04 *
CD146	5 D	40.38 ± 0.53	41.34 ± 1.29	32.41 ± 2.16 **	30.19 ± 3.24 **	36.59 ± 4.24
	10 D	5.54 ± 2.31	3.59 ± 1.25	1.20 ± 0.46 *	0.77 ± 0.13 *	0.99 ± 0.40 *

* *p* < 0.05, ** *p* < 0.01, *** *p* < 0.005.

## Data Availability

The data generated and analyzed during this study is available from the corresponding author on reasonable request.

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
