# Peer review of "Increasing Odontoblast-like Differentiation from Dental Pulp Stem Cells through Increase of β-Catenin/p-GSK-3β Expression by Low-Frequency Electromagnetic Field"

_biomedicines, 2021, doi:10.3390/biomedicines9081049_

Round 1

Reviewer 1 Report

The research work reports the effect of low frequency electromagnetic fields on the differentiation process of pulp stem cells into odontoblasts. Differentiation was evaluated by analyzing odontoblast-related markers by  Western blot analysis, RT-PCR , FACS, and immunofluorescence. Overall, in my opinion, the research work is nicely structured and well defined.
 I expressed the following major issue that needs to be addressed before the paper meets the quality standard for publication.
The authors evaluated the differentiation status of cells by determining the expression of some marker genes using the conventional method of RT-PCR. However, I believe that conventional RT-PCR protocol such as the one used by the authors is not to be considered suitable for the quantitative analysis of genes. For instance, the comparisons are referred to a "constitutive" gene, which is expected but not demonstrated not to change under the experimental conditions because of its lack of reactivity, unrelatedness to the pathways studied, or experimentally observed resilience to change. Moreover, It is not reported the selection of the appropriate number of cycles so that the amplification product is clearly visible on an agarose gel but also so that amplification is in the exponential range and has not reached a plateau yet. The use of RT-PCR in the literature is reported for semi-quantitative analysis only when taking into account some precautions in the experimental protocol that have not been reported by the authors. A semi-quantitative evaluation of gene expression by RT PCR is described in Marone et al., Biol. Proced. Online 2001; 3 (1): 19-25 and Chen et al., Brain Research Protocols 4 1999 132–139
Authors should confirm gene expression data by real time PCR or by demonstrating that they have applied a suitable semiquantitative RT-PCR protocol.

Minor issue:

The statistical differences were not demonstrated/reported for FACS analysis. Please report them in the text.

Author Response

The research work reports the effect of low frequency electromagnetic fields on the differentiation process of pulp stem cells into odontoblasts. Differentiation was evaluated by analyzing odontoblast-related markers by Western blot analysis, RT-PCR , FACS, and immunofluorescence. Overall, in my opinion, the research work is nicely structured and well defined.
 I expressed the following major issue that needs to be addressed before the paper meets the quality standard for publication.
The authors evaluated the differentiation status of cells by determining the expression of some marker genes using the conventional method of RT-PCR. However, I believe that conventional RT-PCR protocol such as the one used by the authors is not to be considered suitable for the quantitative analysis of genes. For instance, the comparisons are referred to a "constitutive" gene, which is expected but not demonstrated not to change under the experimental conditions because of its lack of reactivity, unrelatedness to the pathways studied, or experimentally observed resilience to change. Moreover, It is not reported the selection of the appropriate number of cycles so that the amplification product is clearly visible on an agarose gel but also so that amplification is in the exponential range and has not reached a plateau yet. The use of RT-PCR in the literature is reported for semi-quantitative analysis only when taking into account some precautions in the experimental protocol that have not been reported by the authors. A semi-quantitative evaluation of gene expression by RT PCR is described in Marone et al., Biol. Proced. Online 2001; 3 (1): 19-25 and Chen et al., Brain Research Protocols 4 1999 132–139
1. Authors should confirm gene expression data by real time PCR or by demonstrating that they have applied a suitable semiquantitative RT-PCR protocol.

Ans: Thank you very much for your good comments and point out. We understood your opinion. The synthesized DNA of the last experiment was exhausted. So, we are preparing on cell culture to perform real-time PCR as your comments. Therefore, if you allow more two weeks, we can add the real-time PCR results. Please allow 2 weeks for the 2nd review periods.

  1. Minor issue:

The statistical differences were not demonstrated/reported for FACS analysis. Please report them in the text.

Ans: Thank you very much for your good comments. So, we performed anova statistical analysisas your comment. And we attached raw data as supplementary data.

Table 2. FACS analysis of mesenchymal stem cell markers. FACS data percentage—upper values in each row; 5 days of PEMF exposure, lower values in each row; 10 days of PEMF exposure. Cont = control (no PEMF treatment).

Markers%

Cont

40 Hz

60 Hz

70 Hz

150 Hz

CD73

5 D

10 D

77.21±1.47

7.95±0.34

77.50±4.72

7.50±2.14

61.55±1.57***

7.27±3.29

61.29±0.32***

5.50±1.97

69.25±2.71*

4.56±2.12

CD105

5 D

10 D

43.46±1.09

4.19±0.49

42.33±2.87

3.66±1.22

29.24±6.51*

2.82±1.07

29.33±4.34*

3.38±0.92

36.14±3.76*

2.23±1.04*

CD146

5 D

10 D

40.38±0.53

5.54±2.31

41.34±1.29

3.59±1.25

32.41±2.16**

1.20±0.46*

30.19±3.24**

0.77±0.13*

36.59±4.24

0.99±0.40*

*P<0.05, **P<0.01, ***P<0.005

Reviewer 2 Report

This manuscript entitled “Increasing of Odontoblastic-like Differentiation from Dental Pulp Stem Cells through Increasing of β-catenin/p-GSK-3β expression by Low Frequency Electromagnetic field” is an interesting paper presenting a new protocol for EMF stimulation on DPSC cell culture. The Authors show that a short exposition to EMF compared to previously reported in the literature can increase the expression of odontoblast markers in DPSC culture.

This manuscript is in general well written but should be improved before further consideration for publication.

Comments:

Introduction line 4, the odontoblasts do not by themselves protect from external stimuli. The sentence should be rephrased as : odontoblasts via the production of dentin protect…

The authors are writing odontoblast throughout the introduction. However the work they are referring to evaluated differentiation of DPSC, MSC, and adipose derived MSCs in vitro. There is no guaranty the differentiated cells are true odontoblast, therefore the authors should write odontoblast like cells rather than odontoblast.

In the sentence “Electromagnetic fields (EMFs) affect cell activity, proliferation, differentiation, and proliferation by affecting “  (3rd paragraph of the introduction) there is duplication of “proliferation”.

Last sentence of the introduction is incorrect; the authors didn’t evaluate tooth regeneration but abilities of DPSC to differentiate into odontoblast like cells.

There are inconsistencies on the protocol of EMF exposition. In methods 2.6 for FACS, the cells were differentiated (with EMF exposition? ) for 2 weeks but in the corresponding figure (fig 3) the time points for FACS analysis are day 5 and day 10.

Moreover the methods  2.2 describe that EMF stimulation was 15 min/day for 3 days. But some experiments where performed at D5 and D10 in these experiments was the exposure to EMF 3 days or longer (5 and 10 days)?

A clarification of the protocol of EMF exposure should be added for each experiment.

Why immunostainings where done at day 5 and not D3 like the other assay?

The authors say they performed experiments independently in triplicate. Including several replicate for an experiment means that the sample was used several time in the exact same experimental condition, (same day…) and therefore replicates are not independent. This should be clarified.

Using student’s t test for data analysis is not relevant. In the experiments there is 1 variable and multiple groups so a one way anova followed by tukey’s test (or the corresponding non parametric test if more relevant) should have been done. The statistical analysis for FACS is missing.

Results 3.1 and figure 2 indicate that LDH assay was used to evaluate cell stress. It is incorrect. LDH assay evaluates cell death.

Figure 5, for BMP expression the effects is higher with 150Hz than with 70Hz but not statistically significant, same comment for DSPP and runx2 expression. It would be good to include detailed statistical data (exact p value…) in the results to help the reader to understand this discrepancy.

In their manuscript the authors conclude their protocol of EMF exposition enhance DPSC differentiation into odontoblast like cells. However, the experiments evaluated only the expression of protein and mRNA. The increase in the expression of the markers in EMF groups compared to control may not result in an enhancement of the mineralizing properties of the cells, which is a key endpoint to characterize odontoblast like cells.

A functional assay such as von Kossa or alizarin staining evaluating the formation of calcium nodules should be included in the manuscript.

Author Response

This manuscript entitled “Increasing of Odontoblastic-like Differentiation from Dental Pulp Stem Cells through Increasing of β-catenin/p-GSK-3β expression by Low Frequency Electromagnetic field” is an interesting paper presenting a new protocol for EMF stimulation on DPSC cell culture. The Authors show that a short exposition to EMF compared to previously reported in the literature can increase the expression of odontoblast markers in DPSC culture.

This manuscript is in general well written but should be improved before further consideration for publication.

 Comments:

  1. Introduction line 4, the odontoblasts do not by themselves protect from external stimuli. The sentence should be rephrased as : odontoblasts via the production of dentin protect…

 Ans: Thank you very much for your good point out. I revised this sentence like your comments.

⇒ The odontoblasts secrete extracellular matrix that components of dentin, which acts as barriers to dental pulp protection from external stimuli.

  1. The authors are writing odontoblast throughout the introduction. However the work they are referring to evaluated differentiation of DPSC, MSC, and adipose derived MSCs in vitro. There is no guaranty the differentiated cells are true odontoblast, therefore the authors should write odontoblast like cells rather than odontoblast.

Ans: Thank you very much for your good comment. I corrected word as your comments.

⇒ Substantial research has been conducted on odontoblast differentiation. A comparative study revealed the differentiation of DPSCs, bone marrow-derived mesenchymal stem cells (BM-MSCs), and adipose-derived MSCs into odontoblast-like cells using differentiation medium [16]. Most researchers performed studies using growth factors and cytokines. Kidwai et al. [17] and Ozeki et al. [18] differentiated embryonic stem cells into odontoblast-like cells using bone morphogenetic protein-4. And Jiang et al. induced the differentiation of cranial neural crest cells into odontoblast-like cells by fibroblast growth factor 8 and adult extracellular matrix proteins [17]. In another study, BM-MSCs and amniotic MSCs were differentiated into odontoblast-like cells using a porous scaffold and mechanical stimulation [19]. Those authors applied 19.6 kPa of mechanical compression on the micro-porous scaffold for 9 h, which increased the expression of odontoblast-related markers [18]. Likewise, in a study by Liao et al., mechanical tension induced odontogenic-like differentiation of hDPSCs through the STAT3 signaling pathway by down-regulating sclerostin expression and accelerated senescence of dental pulp cells [20]. Another study promoted odontogenic-like differentiation of BM-MSCs by co-culture with oral epithelial cells [21]. The differentiation-induced cells showed increased expression of DMP-1 and DSPP, and when transplanted in a model rat with dentin defect, a dentin-like structure was regenerated [19].

Jazayeri et al. showed that 15 Hz and 0.2 mT of EMF exposure for 6 h/day induced differentiation of rat MSCs to osteoblast-like cells [30].

Discussion

The purpose of this study was to identify the effects of various frequencies of PEMF on hDPSCs and also determine the optimal condition that would promote their differentiation into odontoblast-like cells.

DMP-1 and DSPP play key roles in dentin mineralization and maturation as major markers of odontoblasts and have been previously used to determine odontoblastic-like differentiation [12, 14, 55-57]

Yun et al. differentiated hDPSCs into odontoblast-like cells using a differentiation medium on a nanofiber scaffold and confirmed to the activity of p38 MAPK signaling by analyzing the protein expression. The results confirmed that p38 MAPK signaling was activated, and p-GSK-3β was increased when hDPSCs were differentiated into odontoblast-like cells [70].

Conclusion

Based on the results, the characteristics of odontoblasts were well expressed after exposure to 70 Hz PEMF for 15 min. Consequently, we have demonstrated for the first time that hDPSCs can effectively differentiate into odontoblast-like cells when exposed to PEMF.

  1. In the sentence “Electromagnetic fields (EMFs) affect cell activity, proliferation, differentiation, and proliferation by affecting “  (3rd paragraph of the introduction) there is duplication of “proliferation”.

 Ans: Thank you very much for your good point out. I removed word as your point out.

⇒ Electromagnetic fields (EMFs) affect cell activity, proliferation, and differentiation by affecting cell membranes or intracellular proteins, such as ion channels [21, 22].

  1. Last sentence of the introduction is incorrect; the authors didn’t evaluate tooth regeneration but abilities of DPSC to differentiate into odontoblast like cells.

Ans: Thank you very much for your good comment. I changed this sentense as your comments.

⇒ We verified the effects of PEMF by analyzing the gene and protein expression levels of hDPSCs exposed to PEMF at different frequencies to find changes indicative of abilities of DPSC to differentiate into odontoblast like cells.

  1. There are inconsistencies on the protocol of EMF exposition. In methods 2.6 for FACS, the cells were differentiated (with EMF exposition?) for 2 weeks but in the corresponding figure (fig 3) the time points for FACS analysis are day 5 and day 10.

Ans: I am sorry about it. It is my mistake. I corrected the experiment period as your indicate

⇒ For analyzing differentiation, hDPSCs were cultured in the same condition as described above (section 2.2 and 2.3) for 10 days.

  1. Moreover the methods  2.2 describe that EMF stimulation was 15 min/day for 3 days. But some experiments where performed at D5 and D10 in these experiments was the exposure to EMF 3 days or longer (5 and 10 days)?

A clarification of the protocol of EMF exposure should be added for each experiment.

Ans: Thank you very much for your comment. I added the experiment period for each experiment as your indicate.

2.4. Cell Viability

For evaluate of cell viability, MTT analysis was performed after 3 days culture.

2.5. Cytotoxicity Assay

The MTT analysis was performed after 3 days culture

2.7. Western Blotting

             For evaluate of protein expression, western blotting was performed after 3 days culture.

2.8. RT-PCR

             For evaluate of mRNA expression, RT-PCR and real time PCR analysis were performed after 3 days culture.

2.9. Immunofluorescence

For evaluate of antigen expression, immunofluorescence staining was performed after 5 days culture

  1. Why immunostainings where done at day 5 and not D3 like the other assay?

 Ans: Thank you very much for your good question.

We have been conducted various studies on neural regeneration, osteogenesis and melanin synthesis using electromagnetic fields. Previous our study of electromagnetic fields, we were evaluated and secured the data for cell activity, mRNA expression, and protein expression after 3 to 7 days culture. In evaluating the effects of electromagnetic fields on cells, the difference between mRNA and protein expression can be seen after 3 days culture, however, surface antigen and morphological differences in cells could not be observed until at least five days of cell culture had passed.

Therefore, in this study, the experimental period was planned differently depending on goal and marker of analysis.

  1. The authors say they performed experiments independently in triplicate. Including several replicate for an experiment means that the sample was used several time in the exact same experimental condition, (same day…) and therefore replicates are not independent. This should be clarified.

  Ans: Thank you very much for your good advices. So, I corrected this sentence.

⇒ All experiments were performed in triplicate.

  1. Using student’s t test for data analysis is not relevant. In the experiments there is 1 variable and multiple groups so a one way anova followed by tukey’s test (or the corresponding non parametric test if more relevant) should have been done. The statistical analysis for FACS is missing.

Ans: Thank you very much for your good comments. So, we performed anova statistical analysisas your comment And I attached raw data as supplementary data. Also, Wee performed statistical analysis for FACS.

⇒  Data are expressed as the mean ± SEM of three independent experiments. One-way analysis of variance (ANOVA) followed by Tukey-Kramer multiple comparisons test was performed with GraphPad Prism (La Kolla, California, USA). Mean differences were considered significant at P < 0.05 (*P < 0.05, **P < 0.01, and ***P < 0.005). Graphical representations were created using SigmaPlot (Systat Software, Inc., San Jose, CA, USA). All experiments were performed in triplicate.

Table 2. FACS analysis of mesenchymal stem cell markers. FACS data percentage—upper values in each row; 5 days of PEMF exposure, lower values in each row; 10 days of PEMF exposure. Cont = control (no PEMF treatment).

Markers%

Cont

40 Hz

60 Hz

70 Hz

150 Hz

CD73

5 D

10 D

77.21±1.47

7.95±0.34

77.50±4.72

7.50±2.14

61.55±1.57***

7.27±3.29

61.29±0.32***

5.50±1.97

69.25±2.71*

4.56±2.12

CD105

5 D

10 D

43.46±1.09

4.19±0.49

42.33±2.87

3.66±1.22

29.24±6.51*

2.82±1.07

29.33±4.34*

3.38±0.92

36.14±3.76*

2.23±1.04*

CD146

5 D

10 D

40.38±0.53

5.54±2.31

41.34±1.29

3.59±1.25

32.41±2.16**

1.20±0.46*

30.19±3.24**

0.77±0.13*

36.59±4.24

0.99±0.40*

*P<0.05, **P<0.01, ***P<0.005

  1. Results 3.1 and figure 2 indicate that LDH assay was used to evaluate cell stress. It is incorrect. LDH assay evaluates cell death.

 Ans: Thank you for your good advice. I corrected word as your advice.

⇒ Furthermore, the absence of significant differences in the amount of LDH released between the groups exposed to PEMF and the controls confirmed that PEMF did not induce cellular death

  1. Figure 5, for BMP expression the effects is higher with 150Hz than with 70Hz but not statistically significant, same comment for DSPP and runx2 expression. It would be good to include detailed statistical data (exact p value…) in the results to help the reader to understand this discrepancy.

 Ans: Thank you very much for your good advices. So, we added the explain about your advices.

⇒ The BMP, OMD, Runx2, and DSPP expression have slight differences in frequency, but not statistically significant. But, the frequency of 60 and 70 Hz showed a high expression in ALP and DMP-1 markers compared to the other frequency statistically significant.

  1. In their manuscript the authors conclude their protocol of EMF exposition enhance DPSC differentiation into odontoblast like cells. However, the experiments evaluated only the expression of protein and mRNA. The increase in the expression of the markers in EMF groups compared to control may not result in an enhancement of the mineralizing properties of the cells, which is a key endpoint to characterize odontoblast like cells. A functional assay such as von Kossa or alizarin staining evaluating the formation of calcium nodules should be included in the manuscript.

Ans: Thank you very much for your good advices. We are preparing on cell culture to perform von Kossa or alizarin staining as your comments. A minimum culture period of 10 days is needed for cells to calm of minerals. And I need about 2 days to staining. So, I think the modification period of 1 week is short.

Therefore, if you allow more two weeks, we can add the staining result. Please allow 2 weeks for the 2nd review periods.

Round 2

Reviewer 1 Report

The authors requested two more weeks to perform the real-time PCR analysis. In my opinion, these data can constitute an improvement in the scientific quality of the manuscript. I leave it to the editor to decide to allow another two weeks for the review work to be completed

Author Response

  1. Authors should confirm gene expression data by real time PCR or by demonstrating that they have applied a suitable semiquantitative RT-PCR protocol.

Ans: Thank you very much for your good comments and point out. We are preparing on cell culture to perform real-time PCR as your comments. As you advised, the results of the experiment were different. There was no statistical difference between the experimental groups. So I attached the result graph that was drawn automatically on the real-time PCR device. I still don't know the exact cause. However, as your advice, we will proceed with the real-time PCR.

2.8. RT-PCR and real-time PCR Analysis

Also, for the real-time PCR analysis, reverse transcriptase reactions were used to synthesize cDNA from 2 µg of total RNA using a dyne RT Dry MIX (Dyne Bio, Gyeonggi-do, Korea) by following the manufacturer's protocols. qPCR was performed using the TB Green® Premix Ex Taq™ (Tli RNaseH Plus) and a StepOnePlus Real-Time PCR System (Applied Biosystems, Waltham, MA) with 60 cycles of denaturation at 95˚C for 5 sec and annealing at 60˚C for 25 sec and amplification at 72˚C for 30 sec. Test gene expressions were normalized against GAPDH in each sample.

  1. Results

As shown in Fig. 5A, all odontogenic genes (bone morphogenetic protein [BMP] 2, ALP, Runx2, osteomodulin [OMD], DMP-1, and DSPP) were expressed more in PEMF-exposed cells than in the controls based on reverse transcriptase-polymer chain reaction (RT-PCR) analysis. The BMP, OMD, Runx2, and DSPP expression have slight differences in frequency, but not statistically significant. But, the frequency of 60 and 70 Hz showed a high expression in ALP and DMP-1 markers compared to the other frequency statistically significant.

However, real-time PCR results differ from RT-PCR. For BMP, OMD, ALP, and Runx2 expression, there was a slight increase in the EMP-exposure groups compared to the non-exposure group, but there were no statistical differences. In particular, DSPP showed a slight decrease in expression compared to non-exposure group groups but increased expression at 150 Hz (Fig. 5B). As a results, no statistical differences were observed in the expression of odontoblast related mRNA by EMF-exposure.

  1. Minor issue:

The statistical differences were not demonstrated/reported for FACS analysis. Please report them in the text.

Ans: Thank you very much for your good comments. So, we performed anova statistical analysisas your comment. And we attached raw data as supplementary data.

Table 2. FACS analysis of mesenchymal stem cell markers. FACS data percentage—upper values in each row; 5 days of PEMF exposure, lower values in each row; 10 days of PEMF exposure. Cont = control (no PEMF treatment).

Markers%

Cont

40 Hz

60 Hz

70 Hz

150 Hz

CD73

5 D

10 D

77.21±1.47

7.95±0.34

77.50±4.72

7.50±2.14

61.55±1.57***

7.27±3.29

61.29±0.32***

5.50±1.97

69.25±2.71*

4.56±2.12

CD105

5 D

10 D

43.46±1.09

4.19±0.49

42.33±2.87

3.66±1.22

29.24±6.51*

2.82±1.07

29.33±4.34*

3.38±0.92

36.14±3.76*

2.23±1.04*

CD146

5 D

10 D

40.38±0.53

5.54±2.31

41.34±1.29

3.59±1.25

32.41±2.16**

1.20±0.46*

30.19±3.24**

0.77±0.13*

36.59±4.24

0.99±0.40*

*P<0.05, **P<0.01, ***P<0.005

Reviewer 2 Report

The authors have improved the manuscript, however, the experiment evaluating calcium nodule formation have not been done.

This experiment could greatly improve the manuscript if the authors could get 2 more weeks for the revision of the manuscript.

Author Response

  1. In their manuscript the authors conclude their protocol of EMF exposition enhance DPSC differentiation into odontoblast like cells. However, the experiments evaluated only the expression of protein and mRNA. The increase in the expression of the markers in EMF groups compared to control may not result in an enhancement of the mineralizing properties of the cells, which is a key endpoint to characterize odontoblast like cells. A functional assay such as von Kossa or alizarin staining evaluating the formation of calcium nodules should be included in the manuscript.

Ans: Thank you very much for your good advices. We are preparing on cell culture to perform von Kossa staining as your comments. However, only 10 days of 2nd revision were given this time. Therefore, we started cultivating, but it was only available for 10 days. Although there was no significant difference between the experimental groups due to the short culture period, faster mineral deposition could be observed in the electromagnetic field exposure groups.

2.10. Von-Kossa Staining

For evaluate of calcium deposition, von-Kossa staining was performed after 10 days culture. The cells was assessed using 5% silver nitrate (Sigma‐Aldrich) solution under ultraviolet light for 20 min, followed by 5% sodium thiosulphate (Sigma‐Aldrich) solution for 5 min, and then counterstained with nuclear fast red solution (Sigma‐Aldrich) for 10 min. The mineral was stained black.

3.6. Increased of Calcium Deposition by PEMF Exposure

The von Kossa staining performed for the evaluation of the calcium deposition of the hDPSC during differentiation (Fig. 8). In growth media culture group, no mineral deposited in the cells surface was observed at all. And the EMF‐exposed groups exhibited a small amount of matrix mineralization, while the Non-exposed (control) group exhibited few matrix mineralization compared with the other groups on Day 10. Not enough mineral deposition was observed due to the short culture period of the cells. However, mineral deposition is formed along the cytoplasm and is observed on the cell surface. At above 60 Hz of frequency, the minor deposition difference was not identified.

Figure 8. Effects of PEMF exposure on calcium deposition as determined by von-kossa staining at 10 days. Mineral detections were increased in PEMF-stimulated cells. PEMF; pulsed electromagnetic field, hDPSC; human dental pulp stem cell, Cont = control (no PEMF treatment). Growth = growth media (no PEMF treatment). Original magnification : ×200, bar = 100 μm

Round 3

Reviewer 1 Report

The authors have adequately responded to my requests for major revisions

Reviewer 2 Report

I have no more comments the manuscript can be published  in the present form